



# Improved Multimodel Superensemble Forecast for Sea Ice Thickness using Global Climate Models

Wang Yangjun1, Liu Kefeng1*, Zhang Ren1, Qian Longxia2, Zhang Yu3

1 Institute of Meteorology and Oceanology, National University of Defense Technology, Nanjing 211101, China

5    2 School of Science, Nanjing University of Posts and Telecommunications, Nanjing 210023, China

PLA 95746 troops, Sichuan 611530, China

*Correspondence to*: Liu Kefeng (lkfnudt@sina.com)

**Abstract.** This paper aims to find an ensemble method that combines the global climate models, providing an accurate forecast of sea ice thickness (SIT). The conventional multimodel superensemble method is widely used in the atmospheric, oceanic,

and other fields, but it does not perform well in SIT simulations. Hence, an adaptive forecasting through exponential re-weighting algorithm is adopted to improve the conventional multimodel superensemble method. The results demonstrate, through a multi-criteria evaluation, that our proposed method performs better than any other mainstream ensemble method. The proposed method is used to predict future SIT in 2020–2049, and its potential biases are discussed.

## 1 Introduction

Since the 1980s, global climate models (GCMs) have developed rapidly with enormous improvements in computer technologies and global climate observation systems, which are important tools for climate predictions (Somerville, 2011). For enhanced data sharing and model comparison, the fifth phase of the Coupled Model Intercomparison Project (CMIP5) has been developed to gather the most advanced GCMs for Arctic sea ice simulations (Taylor et al., 2012). The results show that Arctic sea ice will likely melt and thin in the 21st century (Taylor et al., 2012). However, there are still large uncertainties in

GCM future sea ice thickness (SIT) simulations (Shu et al., 2015; Stroeve et al., 2014) that are determined by the initial conditions, physical processes, and resolutions (Taylor et al., 2012). Thus, future local SIT is predicted at a low level of confidence with a single model (Stephenson and Smith, 2015). These GCMs, albeit with a high uncertainty, can still be applied to the prediction of future Arctic sea ice after careful evaluation (Wang and Overland, 2009).

To improve GCM forecasting accuracy, scholars began to gather the GCM sub candidates into an ensemble forecast, taking

the initial error and model uncertainty into consideration (Hou et al., 2001). The idea was first proposed by Epstein (1969) and Leith (1974), and is now widely used in many countries (Du, 2002). The ensemble mean is the simplest and most commonly used method, demonstrating a performance superior  to that of most single models (e.g. Xu et al., 2010). There is a shortcoming in the ensemble mean method where the models with poor skills carry the same weight as those with high ones.

The bias-removed ensemble mean method (e.g. Zhi et al., 2011; Zhu, 2011), assumes that the poorest model can be made

equivalent to the best model with a bias correction. Then, the bias-removed models can be assigned the same weights.

Krishnamurti et al. (1999) adopted a new method called the multimodel superensemble, which utilises linear regression to

minimise the errors between GCMs and observations at the grid level during the training period, and different GCM weights

are obtained and transferred to the forecast phase. This method can effectively reduce the local biases in space and time and

those of vast parameters on different models, as it is far more particular in its weight assignments compared with that of the

other two methods (Krishnamurti et al., 2016). Existing studies illustrate that the multimodel superensemble has been widely

applied in weather and climate (e.g. Derber and Wu, 1998; Kazumori et al., 2008; Leutbecher, 2003; Mahfouf and Rabier,

2000), ocean (e.g. Kantha et al., 2008; Lenartz et al., 2010a, 2010b), hurricane (e.g. Munsell et al., 2015; Rios-Berrios et al.,

2014; Sato and Xue, 2013; Xue et al., 2013), and other forecasting, significantly reducing the prediction error. Additionally,

ensemble forecasting based on artificial intelligence technology has been gradually developed (e.g. Zhi ying et al., 2004; Gui

and Zhao, 2013; Shi, 2013; Zhang et al., 2018).

However, there is a lack of research on the ensemble forecasting of SIT. Compared with the sea ice concentration (SIC), SIT

is a complex variable that is difficult to observe (Haas, 2010) without sufficient and reliable large-scale satellite observations

(e.g. Laxon et al., 2013; Tilling et al., 2016). Both statistical methods (e.g. Lindsay et al., 2008) and numerical models (e.g.

Holland et al., 2011) show that there is a strong correlation between SIT and sea ice extent (SIE). The change in SIT is more

significant and can provide more information than SIC, especially in central areas (Melia et al., 2015c). The reduction in SIT

extends the navigation season, making high-latitude sailing possible (Smith and Stephenson, 2013) and facilitating the

exploration of abundant natural resources, impacting the Arctic ecosystem and mid-latitude climate (Francis and Vavrus, 2012).

However, experiments have illustrated that the conventional multimodel superensemble based on linear regression cannot

simulate SIT well due to the sparsity of temporal and spatial SIT data. Moreover, Yang (2001) pointed out that complicated

ensemble methods can lead to unstable weights and inferior performances than those of the best candidates. Therefore, an

ensemble method with an improved forecasting capacity is required for further investigations.

To fill this research gap, this study incorporates 12 high-performing GCMs for different scenarios and initial conditions that

were evaluated by Wang and Overland (2015), amounting to 101 ensemble candidates in total, and a new method called

adaptive forecasting through exponential re-weighting (AFTER) is adopted to improve the conventional multimodel

superensemble method. Monthly SIT data from 2006–2017 were used in the training phase, while monthly data from 2018

were used in the test phase. A multi-criteria evaluation, including root mean square error (RMSE), correlation coefficient (CC),

structural similarity index measure (SSIM), empirical orthogonal function (EOF) analysis, and sea ice volume (SIV), is

incorporated in this study to examine the validation of the proposed method and other mainstream ensemble methods, e.g. the

ensemble mean, bias-removed ensemble mean, multimodel superensemble, and artificial neural network. The results show that

the improved multimodel superensemble algorithm has a superior performance to that of the other algorithms. Finally, a new

method is adopted for SIT projection from 2020 to 2049.

The reminder of this paper is organised as follows. A data description is presented in Section 2. Section 3 introduces the

methodology, followed by the model validation test in Section 4. Finally, Section 5 provides the future SIT predictions and

summary.

## 2 Observations and climate simulations

### 2.1 Pan-Arctic Ice-Ocean Modelling and Assimilation System data

Spatial consistency, temporality, and completeness are key factors in data evaluation (Melia et al., 2015a). The Pan-Arctic Ice-

Ocean Modelling and Assimilation System (PIOMAS) sea ice reanalysis data, which assimilated the atmospheric reanalysis

from the National Centres for Environment Prediction, consists of SIC satellite (Lindsay and Zhang, 2006) and sea surface

temperature observations (Schweiger et al., 2011); this dataset was selected for use in this study (Zhang and Rothrock, 2003).

The quality of PIOMAS was evaluated by Schweiger et al. (2011), demonstrating biases in PIOMAS of 0.26 m in autumn and

0.1 m in spring, compared with the ICESat data (Zwally et al., 2002). Although uncertainty exists in the PIOMAS data, current

satellite observations (i.e. ICESat or CryoSat-2) have limited spatial and temporal coverage, restricting their ability to evaluate

models. Moreover, the largest discrepancy between PIOMAS and ICESat data is found in the north of Greenland and the

Canadian Archipelago, the thickest sea ice areas; the PIOMAS data have fewer discrepancies with the situ data than that of

ICESat due to the difference in the satellite inversion methods (Schweiger et al., 2011). Labe et al. (2018) pointed out that the

spatial patterns, seasonal cycles, and SIT trends are sufficiently reproduced by the PIOMAS data. Therefore, PIOMAS has

been widely used to represent observations in various studies (e.g. Shu et al., 2015; Labe et al., 2018).

### 2.2 Global climate models

This study incorporates 12 GCMs from CMIP5 that were evaluated by Wang & Overland (2015) for a combined forecast, with

a total of 101 ensemble candidates for four emission scenarios called representative concentration pathways (RCPs) 2.6, 4.5,

6.0, and 8.5 (van Vuuren et al., 2011).

The basic characteristics of the selected ensemble candidates are displayed in Table 1. Regarding the discrepancies in spatial

resolution, all the model candidates and PIOMAS were interpolated into the same $1° \times 1°$ resolution.

For each candidate of the 12 GCMs, monthly data for 2006–2017 were utilised in the training phase because RCPs were first

used in 2006. Then, monthly data from 2018 were used in the test phase to validate different ensemble methods. Finally,

monthly SIT ensemble data was forecasted for 2019–2050.





**Table 1. List of models used in the CMIP5 subset**

| Number | Model Name | Spatial Resolution | Ensemble candidates (RCP) | | | | Reference |
|---|---|---|---|---|---|---|---|
| | | | 26 | 45 | 60 | 85 | |
| 1 | ACCESS1.0 | tripolar, $1° \times 1°$, refinement at the equator | | 1 | | 1 | (Bi et al., 2013) |
| 2 | ACCESS1.3 | tripolar, $1° \times 1°$, refinement at the equator | | 1 | | 1 | (Bi et al., 2013) |
| 3 | CCSM4 | dipolar, $1.11° \times (0.27 - 0.54)°$, NP in Greenland | 5 | 6 | 6 | 6 | (Gent and Danabasoglu, 2011) |
| 4 | CESM1 | dipolar, $1.11° \times (0.27 - 0.54)°$, NP in Greenland | 3 | | | 1 | (Gent and Danabasoglu, 2011) |
| 5 | EC-EARTH | tripolar, $1° \times 1°$, refinement at the equator | 2 | 10 | | 10 | (Fichefet and Maqueda, 1999) |
| 6 | HadGEM2-ES | $(1 - 0.3)° \times 1°$ | 4 | 4 | 4 | 5 | (Mclaren et al., 2006) |
| 7 | HadGEM2-CC | $(1 - 0.3)° \times 1°$ | 1 | 1 | 1 | 1 | (Mclaren et al., 2006) |
| 8 | HadGEM2-AO | $(1 - 0.3)° \times 1°$ | | 1 | | 3 | (Mclaren et al., 2006) |
| 9 | MIROC-ESM | $\sim 1.4° \times 1°$ | 1 | 1 | 1 | 2 | (Watanabe et al., 2011) |
| 10 | MIROC-ESM-CHEM | $\sim 1.4° \times 1°$ | 1 | 1 | 1 | 1 | (Watanabe et al., 2011) |
| 11 | MPI-ESM-LR | $\sim 1.5° \times 1.5°$ | 3 | 3 | | 3 | (Notz et al., 2013) |
| 12 | MPI-ESM-MR | $\sim 0.4° \times 0.4°$ | 1 | 3 | | 1 | (Notz et al., 2013) |
| Sum | | | 21 | 32 | 13 | 35 | 101 |

**3 Ensemble forecast methodology**

The ensemble forecast aims to improve the model projection accuracy by making full use of multiple information sources from the GCMs and constraining they GCMs using observations. We have tested the performances of different ensemble forecast methods that are seldomly used in SIT projection (Section 4). SIT simulations based on the conventional multimodel superensemble, an advanced method that can significantly improve predictions in other areas, exhibit large observation biases due to the linear regression overfitting. Therefore, a new weight determination method was adopted to improve the

conventional multimodel superensemble. The ideas behind this improved method and other ensemble forecast methods used in this study are introduced in this section. The mathematical notation for the following equations is in Table 2.

**Table 2. Notation key**

| Notation | Description |
|---|---|
| $M_i$ | The ith candidate of the total ensemble candidates $i \in N$ |
| $L$ | Latitude and longitude information of each grid |
| $O_h$ | PIOMAS data over the training period (2006–2017) |
| $x_h$ | $x$ over the training period (2006–2017) |
| $\bar{x}$ | Time mean of $x$ over the training period |
| $\langle x \rangle$ | Ensemble mean of $x$ |
| $\hat{x}$ | Temporally detrended $x$ over the training period |
| $\tilde{x}$ | Temporally trend $x$ over the training period |



| Notation | Description |
|---|---|
| $x_t$ | $t$ over the test period (2018) |
| $\sigma$ | Standard deviation |
| MEAN | Ensemble mean method |
| BIAS | Bias-removed ensemble mean method |
| MARVIC | MARVIC method |
| ANN | Ensemble forecast with ANN |
| SUPER | Conventional multimodel superensemble method |
| AFTER.L1 | Improved multimodel superensemble method with L1-norm AFTER |
| AFTER.L2 | Improved multimodel superensemble method with L2-norm AFTER |

**3.1 Ensemble mean**

The ensemble mean method was widely used in the fifth report from Intergovernmental Panel on Climate Change to predict

atmospheric, oceanic, and cryospheric variables. This approach averages all the ensemble candidates, $\langle M_{i,t} \rangle$, regardless of

model discrepancy.

$$SIT_{MEAN} = \langle M_{i,t} \rangle. \tag{1}$$

**3.2 Bias-removed ensemble mean**

Bias-removed ensemble mean methods attempt to correct the models with poor accuracies before averaging them. The

conventional approach corrects the time mean by subtracting the biases between each ensemble candidate, $\bar{M}_{i,h}$, and

observation, $\bar{O}_h$, during the training period at the grid level.

$$SIT_{BIAS} = \langle M_{i,t} - (\bar{M}_{i,h} - \bar{O}_h) \rangle. \tag{2}$$

The mean and variance correction ensemble mean (MAVRIC), first proposed by Nathanael Melia et al. (2015), attempts to

consider both mean and variance in the bias correction. This study incorporates MAVRIC using the ratio of the temporal

standard deviation of the detrended observations, $\sigma_{\tilde{O}_h}$, to the standard deviation of each detrended candidate, $\sigma_{\tilde{M}_{i,h}}$, over the

training period (Eq. 3). Each model is detrended using the linear time series trend during the training period. The multiplicative

correction is first detrended, the variance is then corrected, and the trend is re-applied.

$$SIT_{MAVRIC} = \langle (M_{i,t} - \tilde{M}_{i,t}) \frac{\sigma_{\tilde{O}_h}}{\sigma_{\tilde{M}_{i,h}}} + \tilde{M}_{i,t} \frac{\bar{O}_h}{\bar{M}_{i,h}} \rangle. \tag{3}$$

**3.3 Ensemble forecast via artificial neural network**

The ensemble forecast with an artificial neural network (ANN) is biologically motivated, imitating the abilities of the human

brain including information storing, learning, and training to minimise the difference between multi-models and observations.

The algorithm structure can be seen in Eq. (4) and Figure 1. The input layer consists of all the candidates and their geographic

information (latitude and longitude), and each grid with a single time clip during the training period is treated as a training

sample, 905,616 in total with 103 dimensions. The output layer consists of related observations. The network can be obtained

through the training phase and used for predictions.



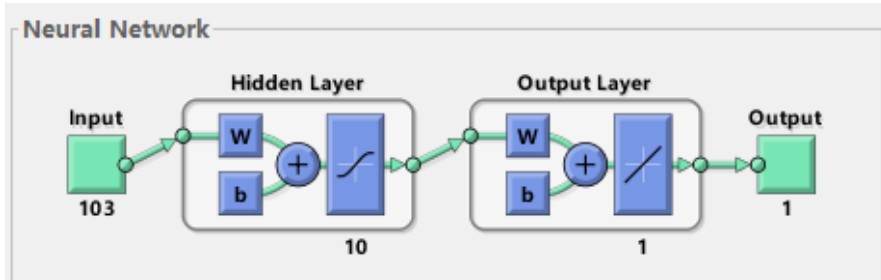

**Figure 1:** ANN network.

$$SIT_{ANN} = net(M_{i,t}, L), \tag{4}$$

### 3.4 Multimodel superensemble forecasting and its improvement

The conventional multimodel superensemble method is a type of regression-improved forecast that provides weights for each grid (Eq. 5). The weights can be obtained by minimising the errors between all the candidates and observations during the training phase, and a linear regression is most commonly used (Eq. 6).

$$SIT_{SUPER} = \overline{O_h} + \langle a_i(M_{i,t} - \overline{M}_{i,h}) \rangle, \tag{5}$$

$$a_i \in argmin \sum_{j=1}^{h}(M_{i,j} - O_j)^2. \tag{6}$$

In our experiment, this superensemble method contributes to large systematic errors in SIT projection (Section 4). As a solution for this, we adopted the AFTER algorithm in the superensemble structure to improve the weight calculations, maintaining all positive weights to avoid overfitting and instability. The improved algorithm was first proposed by Yang (2001b) and can be presented as follows:

$$SIT_{AFTER} = \overline{O_h} + \langle w_i(M_{i,t} - \overline{M}_{i,h}) \rangle, \tag{7}$$

$$w_i = \frac{E_i}{\sum_{j=1}^{n} E_j}, \tag{8}$$

$$E_i = \prod_{k=1}^{n} \hat{s}_{i,k}^{-n/2} \exp(-\lambda L_i). \tag{9}$$

Two weighting forms are proposed, and the L1-norm AFTER algorithm is as follows.

Step 1. Split the data of each candidate for the training period in two parts, $Z^{(1)} = (M_{i,h1}, O_{i,h1})$, $1 \le h_1 \le n/2$ and $Z^{(2)} = (M_{i,h2}, O_{i,h2})$, $n/2 + 1 \le h_2 \le n$.

Step 2. Based on $Z^{(1)}$, compute the mean absolute prediction error $\hat{s}_{k,j} = (2/n) \sum_{h_1=1}^{n/2} |M_{i,h1} - O_{h1}|$ for the $ith$ candidate model.

Step 3. For the $ith$ model, the loss function, $L_i$, from Eq. 9 can be written as

$$L_i = \sum_{h2=n/2+1}^{n} |M_{i,h2} - O_{h2}|. \tag{10}$$

Step 4. Compute the convex weight for the $ith$ model following wed by Eqs. (8) and (9).

Step 5. Randomly permute the order of the data N-1 times. Repeat Steps 2–5.

Note that a tuning parameter $\lambda$ is used to control the effect of weighting on the forecast performance (normally, $\lambda = 1$).





The workflow of the L2-norm AFTER algorithm is similar to that of L1-norm AFTER, expect for the function in Step 3, which should be rewritten as

$$L_i = \sum_{h2=n/2+1}^{n} \frac{(M_{i,h2}-O_{h2})^2}{2\hat{s}_{h2}^2}. \tag{11}$$

**4 Method validation**

In this study, we analysed the performance of the ensemble forecast methods mentioned above using a statistical multi-criteria evaluation approach. Both univariate and multivariate techniques including RMSE, CC, SSIM, and EOF analysis are adopted to capture the reliability and nature of the ensemble models. The analysis is performed both spatially and temporally on the ensemble forecast datasets during the testing phase. The temporal scale analysis was used to understand the prediction ability of different ensemble forecast methods for the SIT variation trends, testing whether the methods can perform well after being

fully trained. Gridded data were used to analyse the discrepancies between the diverse ensemble forecast models in different regions. Finally, these ensemble datasets were adopted to reproduce the monthly variations of the SIV in 2018. The results have provided various statistical properties for these methods.

**4.1 RMSE & CC test**

The RMSE of the datasets measures the deviation between the simulation and observation. The CC of the datasets refers to

the degree of linear correlation between them, combining the concepts of mean, standard deviation, and regression line. In this study, both the RMSE and CC of different SIT ensemble forecasts were calculated using the spatial (Figures 2 and 4) and temporal means (Figures 3 and 5).

Figure 2 illustrates that the two improved multimodel superensemble methods have the minimum spatial average RMSEs, during each month in the testing phase, showing the least amount of bias compared with the observation trends. In this

experiment, the bias-removed ensemble mean algorithm and ensemble forecast with the ANN algorithm can also perform better than any single ensemble candidate, improving the simulation accuracy of the conventional models. Combining the four algorithms, the largest RMSE occurs in August during the testing period, which is consistent with that of the greatest SIV anomalies driven by a positive feedback loop between the SIT and ice-albedo (Bushuk et al., 2017). That kind of feedback is affected by melt pond information, snowfall, and sea ice concentration, which cannot be sufficiently simulated by the current

GCMs from the CMIP5 (Stroeve et al., 2014), restricting the effectiveness of the ensemble forecast in that month.

Figure 3 reflects the temporally averaged RMSEs between the ensemble models and observations at the grid level. Two superensemble methods modified by AFTER, and the bias-removed ensemble mean method captured less of the RMSE than the other selected methods in most parts of the Arctic region. In view of the spatial distribution, the highest biases arose along

the coastlines, especially in the East Siberian Sea, north of the Canadian Arctic Archipelago, and coastlines of Greenland extending to the islands of Svalbard, the highest thickness variability area (Blanchard-Wrigglesworth and Bitz, 2014).

Figure 4 illustrates that the best three ensemble forecast models based on averaged spatial CC tests are the two improved multimodel superensemble methods and the bias-removed ensemble mean method, where the values are approximately 0.9 for all the months. The latter method performs better from July to November, while our proposed methods have an advantage in the remaining months of 2018, and all of the models together have a higher correlation with the observations than that of any

single ensemble candidate. Moreover, these methods and the ANN method conclude that the least relevant occurs in August, while that appears in September for other methods, providing more evidence for model evaluation. From the spatial distribution perspective, we can see that the datasets from all the listed ensemble methods in Figure 5 are highly correlated with the observation data for most Arctic regions, except for an area in the north of the Canadian Arctic Archipelago, from the Fram Strait to the Norwegian Sea, and part of the Barents Sea, which demonstrated a discrepancy compared to the RMSE test results.

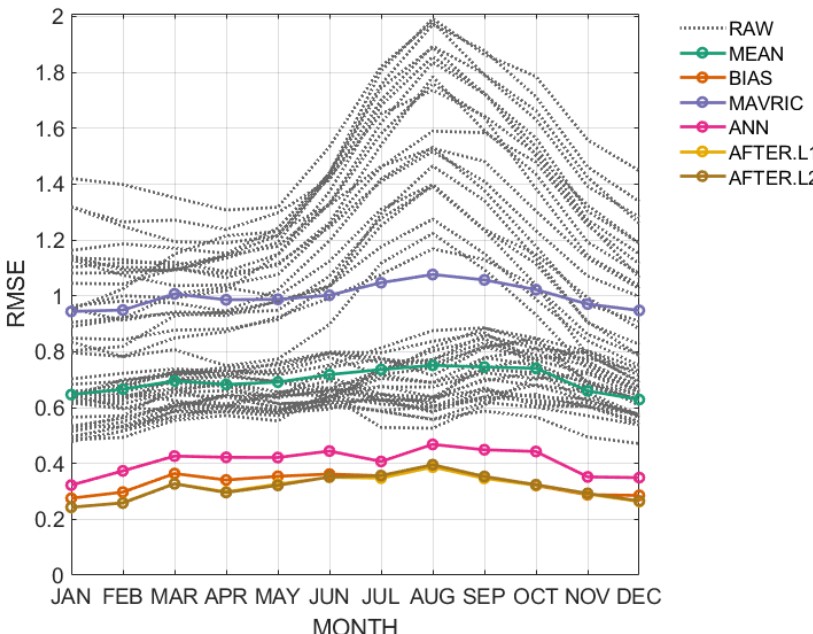


**Figure 2: Spatially averaged SIT RMSEs in multiple datasets based on different ensemble forecast methods and observations in 2018. Results of all 101 candidates are depicted using grey dashed lines, which are marked as raw data. All model abbreviations are the same as those provided in Table 2. Note that the average RMSEs based on the conventional multimodel superensemble method are larger than $10^6$ and are not plotted in this figure.**





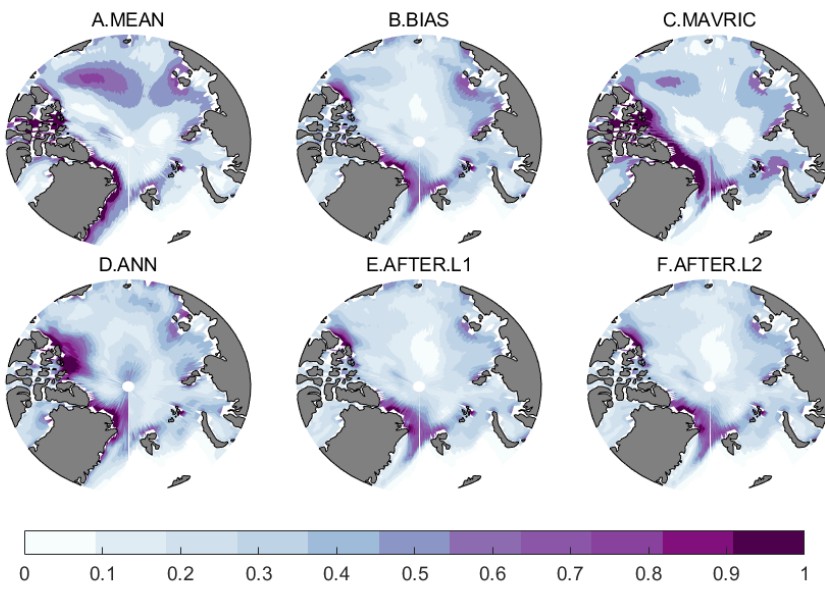

**Figure 3: Temporally averaged SIT RMSEs in multiple datasets are based on different ensemble forecast methods and gridded observations for 2018. All model abbreviations are the same as those provided in Table 2. Note that the average RMSEs based on the conventional multimodel superensemble method are larger than $10^5$ and are not plotted in this figure.**

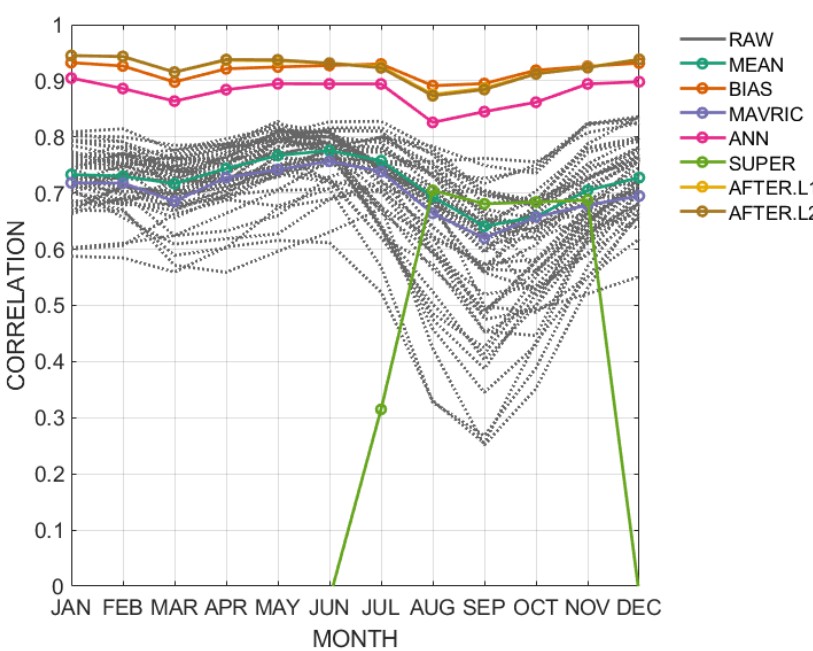


**Figure 4: Spatially averaged SIT CCs for multiple datasets based on different ensemble forecast methods and observations for 2018. Results of all 101 candidates are depicted using grey dashed lines, which are marked as raw data. All model abbreviations are the same as those provided in Table 2.**

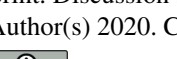



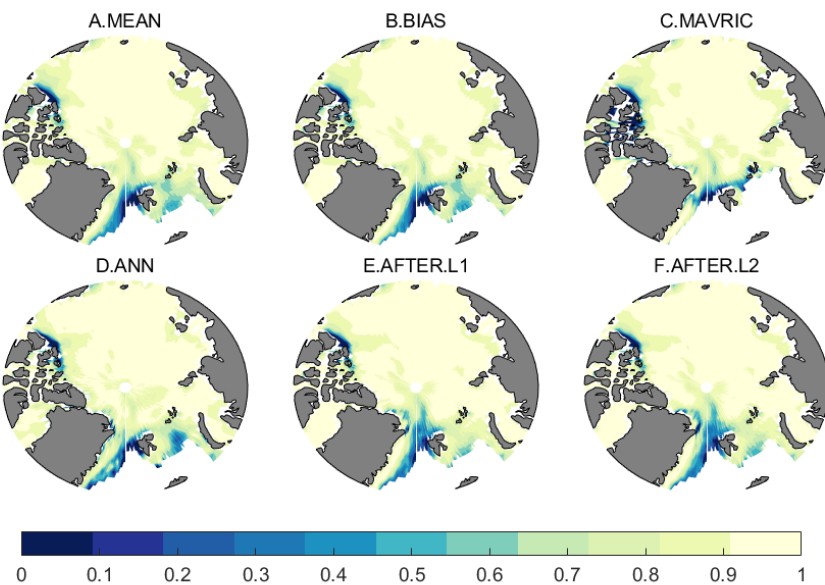

**Figure 5: Temporally averaged SIT CCs in multiple datasets based on different ensemble forecast methods and gridded observations for 2018. All model abbreviations are the same as those provided in Table 2. Note that the average CCs based on the conventional multimodel superensemble method perform poorly and are not plotted in this figure.**

**4.2 Structural similarity index measure (SSIM)**

The SSIM is used to evaluate the performance of different ensemble models by adopting a single index that can measure overall spatial patterns, which is a type of objective test method (Wang et al., 2004). Compared to traditional error methods (e.g. RMSE), this method can better depict the spatial distribution differences between model outputs and observations. The formula of this SSIM can be presented as follows:

$$SSIM(X_{mod}, X_{obs}) = l(X_{mod}, X_{obs})^\alpha \cdot c(X_{mod}, X_{obs})^\beta \cdot s(X_{mod}, X_{obs})^\gamma, \qquad (12)$$

$$l(X_{mod}, X_{obs}) = \frac{\left(2\mu_{x_{mod}}\mu_{x_{obs}} + C_1\right)}{\left(\mu_{x_{mod}}^2 + \mu_{x_{obs}}^2 + C_1\right)}, \qquad (13)$$

$$c(X_{mod}, X_{obs}) = \frac{2\sigma_{x_{mod}}\sigma_{x_{obs}} + C_2}{\sigma_{x_{mod}}^2 + \sigma_{x_{obs}}^2 + C_2}, \qquad (14)$$

$$s(X_{mod}, X_{obs}) = \frac{\sigma_{x_{mod}x_{obs}} + C_3}{\sigma_{x_{mod}}\sigma_{x_{obs}} + C_3}, \qquad (15)$$

where the variations in mean value, $l(X_{mod}, X_{obs})$, deviation, $c(X_{mod}, X_{obs})$, and structure, $s(X_{mod}, X_{obs})$ are combined together; $\mu$ and $\sigma$ are the mean value and standard deviation of $X$, respectively; and $C_1, C_2,$ and $C_3$ are the constants to prevent system instability. Generally, if $\alpha = \beta = \gamma = 1, C_3 = C_2/2$ , then Eq. (12) can be rewritten as follows:

$$SSIM(X_{mod}, X_{obs}) = \frac{\left(2\mu_{x_{mod}}\mu_{x_{obs}} + C_1\right)\left(2\sigma_{x_{mod}}\sigma_{x_{obs}} + C_2\right)}{\left(\mu_{x_{mod}}^2 + \mu_{x_{obs}}^2 + C_1\right)\left(\sigma_{x_{mod}}^2 + \sigma_{x_{obs}}^2 + C_2\right)}. \qquad (16)$$

A sample model with a random matrix A of $10 \times 10$ that ranges from 0 to 1 is provided to verify the advantage of SSIM. Matrix B can be obtained if 10 is added to the last element, while Matrix C can be obtained if we add 1 or -1 randomly to each





element. Both the SSIM and RMSE are calculated to test the difference between B and A, and that between C and A. The

results show that the RMSE of these matrices are the same, while the SSIM calculated for Matrix B is larger than that of Matrix

C, showing that the SSIM provides more information than the RMSE (Figure 6).

Figure 7 illustrates that the improved multimodel superensemble methods have higher scores than those of any other single

candidate every month in the SSIM, matching the spatial distributions with the observations. The improved methods, together

with the bias-removed and ANN methods, demonstrate that the largest SIT spatial distribution biases arise in August, which

the other ensemble methods cannot capture.

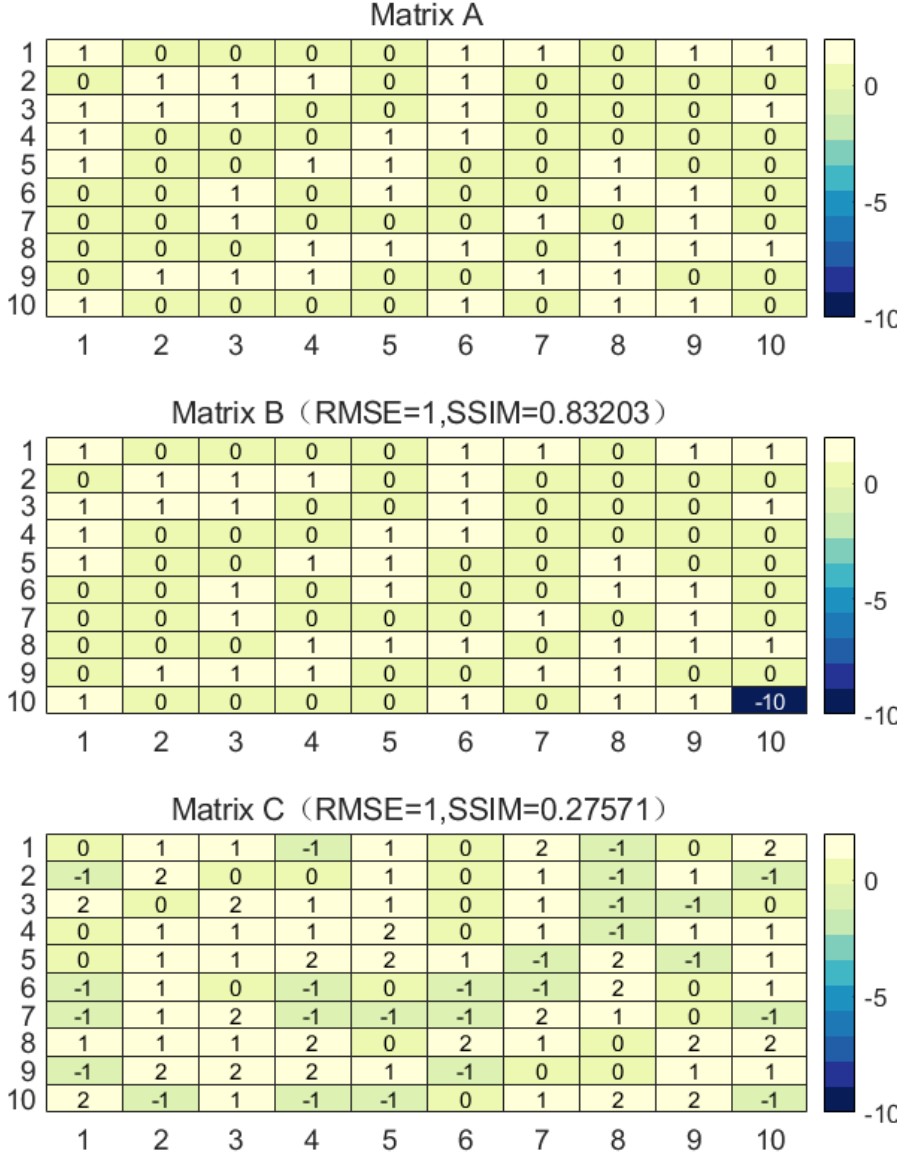


**Figure 6: An example of the structural similarity analysis.**

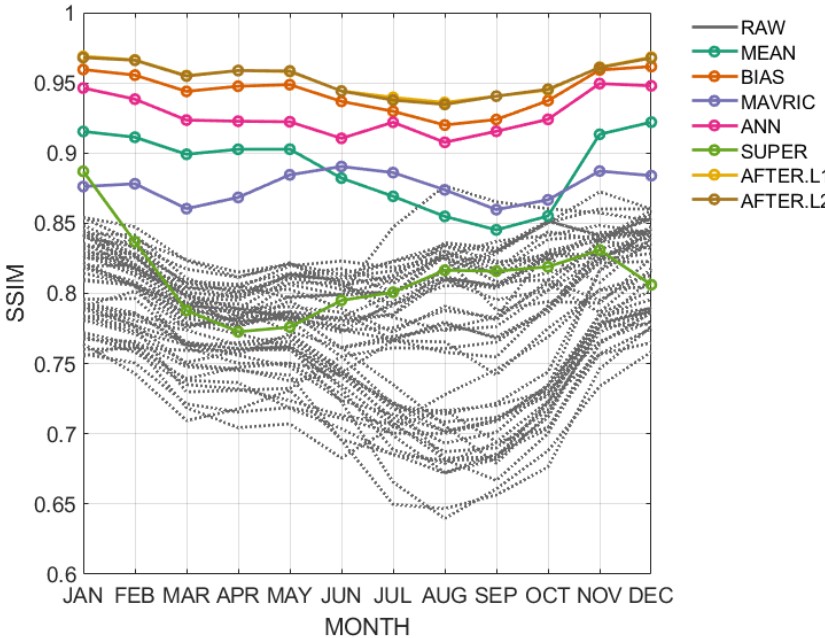

**Figure 7: The SSIM of SIT in multiple datasets based on different ensemble forecast methods and observations in 2018. Results of all 101 candidates are depicted using grey dashed lines and are marked as raw data. All model abbreviations are the same as those provided in Table 2.**

### 4.3 EOF analysis

The EOF analysis is introduced to further evaluate the model performances to find a reliable ensemble model to reproduce a realistic sea ice climatology, consistently capturing the major spatial modes and their related 1D principal components (PC). In this study, raw datasets from seven ensemble models and observations are processed by the EOF analysis, where the cumulative total variance of the first four leading EOF patterns is over 99% (Figures 8 and 9). The first spatial pattern reflects the monthly mean SIT fields during the testing phase, and all the ensemble methods except for the conventional superensemble method have similar EOF1 patterns, with the thickest ice north of Greenland and the Canadian Arctic Archipelago, and thinner ice east of the Arctic Basin. The corresponding PC1 patterns of the improved superensemble forecast methods match those of the observations the best, while others have smaller fluctuations. The EOF2 patterns are similar in most ensemble datasets (except for the pattern from the conventional superensemble method), exhibiting a high positive centre of variation in the northern Arctic Basin and a negative centre north of Greenland. Their corresponding PC patterns in Figure 9 consistently depict the SIT downtrend from February to September and uptrend from June to October. In Figure 8, for each EOF3 pattern, only the bias-removed ensemble mean, ANN, and the two improved superensemble methods match the observations, showing a "positive–negative–positive" spatial structure from the west to the east of the Arctic regions. When combined with their corresponding PC3 patterns in Figure 9, the ANN method becomes the only ensemble model to match the observations. The



EOF4 patterns of most of the datasets exhibit similar structure, leading to disorders in their corresponding PC patterns, while only the ANN method can capture a similar trend and amplitude as that of the observations in the fourth PC pattern, meaning that this method displays more detailed results.



**Figure 8: The first four EOF patterns of the seven ensemble forecast models and the observations. Note: areas in red indicate the highest positive anomalies, while areas in blue indicate the highest negative anomalies.**



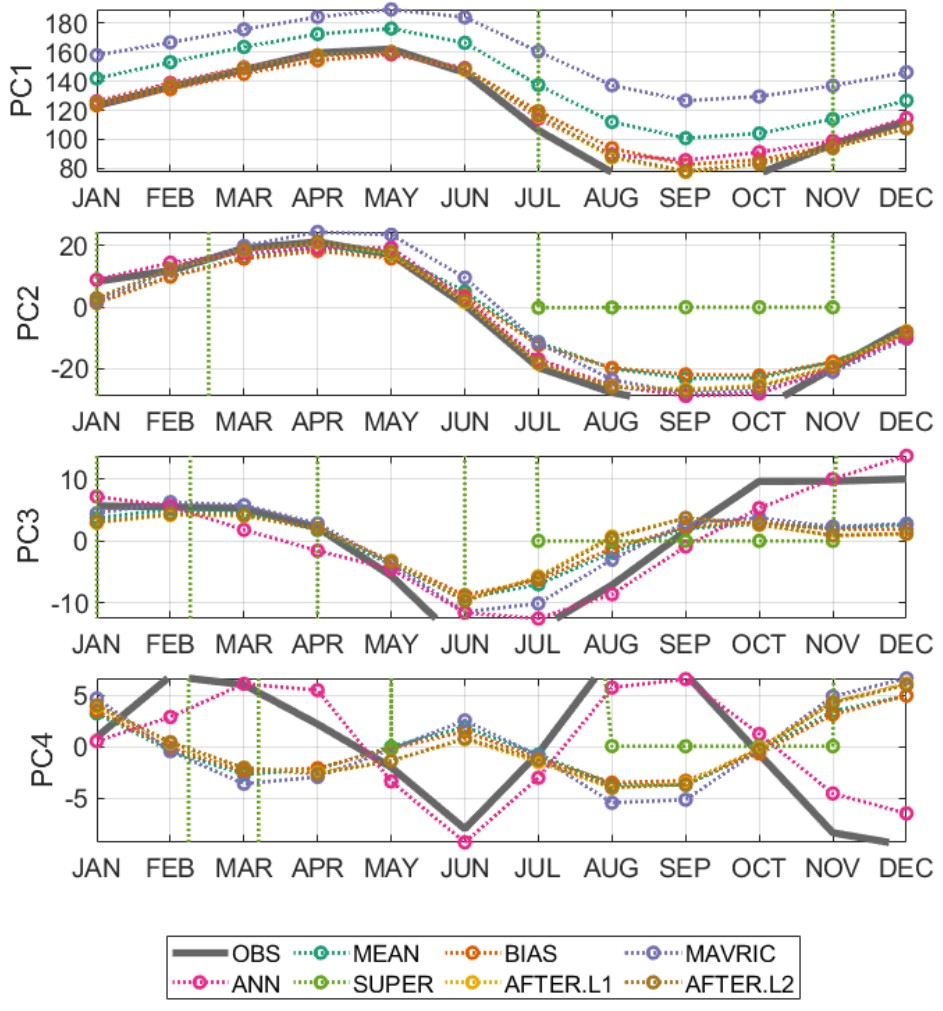

**Figure 9: PC patterns (time coefficients) of the first four leading modes.**

**4.4 Sea ice volume**

Sea ice volume (SIV) is an important index in the assessment of sea ice simulations, combining SIT, SIC, and grid area into

consideration. Here, SIC datasets from PIOMAS and the GCMs listed in Table 2 are adopted to calculate SIV (Eq. 17).

$$SIV = \sum_{lon} \sum_{lat} SIT(lon, lat) \cdot SIC(lon, lat) \cdot \left( sin\left(\frac{lat+1}{180} \cdot \pi\right) - sin\left(\frac{lat}{180} \cdot \pi\right) \cdot \frac{2\pi r^2}{360} \right). \tag{17}$$

Figure 10 illustrates that SIV datasets from the improved superensemble methods have the most similar variations in trend and

amplitude compared to those of the observations. SIV simulations from the other methods, such as the bias-removed ensemble

mean and ANN, are lower than those of the observations in most months, but higher from August to October during the testing

phase.





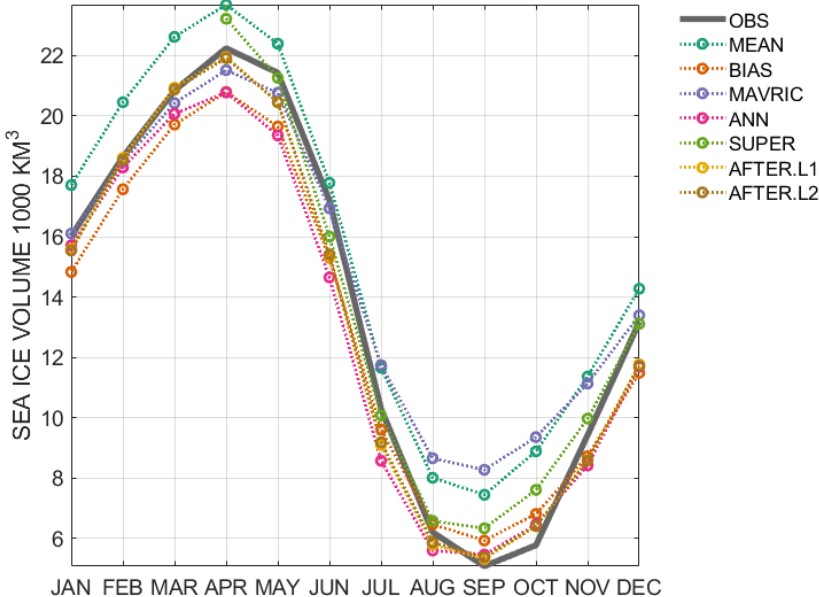

**Figure 10: SIV in multiple datasets based on different ensemble forecast methods and observations for 2018. All model abbreviations are the same as those provided in Table 2.**

### 3.4 Ranking

Ranking is provided for the ensemble methods listed in Table 2 based on a multi-criteria evaluation. The best model is given the top ranking, while the worst model is ranked seventh. Table 3 shows that our proposed methods perform better than the other selected ensemble methods for most evaluation techniques. Additionally, the ANN ensemble method can capture more details of the observation dataset in the EOF analysis owing to the advantage of artificial intelligence in data mining. Even the simplest method, the ensemble mean, is highly correlated with the observation dataset in the spatial distribution. To summarise,

our proposed methods can greatly improve the accuracy of the ensemble forecast in SIT, which is followed by the conventional bias-removed ensemble mean algorithm and ANN ensemble methods, while the remaining three methods score the lowest.



**Table 3.** Performance of ensemble methods as evaluated against the observation dataset for each metric based on spatially averaged and temporally averaged RMSE and CC, the SSIM mean, the absolute discrepancy of EOF of the spatial and PC temporal means, and the standard deviation of SIV.

|  | MEAN | BIAS | MAVRIC | ANN | SUPER | AFTER.L1 | AFTER.L2 |
|---|---|---|---|---|---|---|---|
| RMSE (space) | 0.4040 | 0.2849 | 0.4999 | 0.3470 | $>10^5$ | 0.2690 | 0.2696 |
| RMSE (time) | 0.6968 | 0.3326 | 0.9997 | 0.4065 | $>10^6$ | 0.3136 | 0.3155 |
| CC (space) | 0.8583 | 0.8578 | 0.8447 | 0.8513 | 0.2606 | 0.8597 | 0.8546 |
| CC (time) | 0.7208 | 0.9186 | 0.7001 | 0.8791 | 0.2439 | 0.9226 | 0.9221 |
| SSIM | 0.8894 | 0.9436 | 0.8763 | 0.9275 | 0.8121 | 0.9535 | 0.9531 |
| EOFs (10e-4) | 1.574 | 0.8421 | 2.148 | 0.3165 | 4.210 | 0.5762 | 0.6029 |
| PCs (weight) | 0.8614 | 1.180 | 1.886 | 0.4033 | $>10e6$ | 0.7310 | 0.7378 |
| SIV (std: 10e5) | 1.664 | 1.145 | 1.244 | 1.116 | $>10^5$ | 0.7018 | 0.6979 |
| RANK | 7 | 6 | 5 | 4 | 3 | 2 | 1 |

## 4  Prediction and summary

Based on the study results, the improved superensemble methods are the best ensemble models for simulating SIT, and the improved model with L1-norm AFTER performs better than the other one. Hence, the L1-norm AFTER superensemble method is adopted to predict future variations of the September mean SIT during three periods (2020–2029, 2030–2039, and 2040–2049), where the bias-removed ensemble mean is used as a control group. Figure 11 illustrates that the sea ice will continue to melt in the next three decades and the two selected ensemble models exhibit a similar overall spatial distribution, i.e. thicker sea ice in the west and nearly ice-free in the east.

However, even these two high performing methods still exhibit discernible differences that increase with the prolongation of time. Compared to the results of the bias-removed ensemble mean, the SIT distribution exhibits a "decrease–increase–decrease" belt from the north of Canadian Arctic Archipelago and Greenland Sea to the Barents and Kara Sea in the first decade of the simulation. Then the SIT in the centre of the Arctic Basin increases, while the SIT decrease in the Barents and Kara Sea disappears during 2030–2039. By the middle of the 21st century, the discrepancy arises in the Arctic Basin, while the "two opposite poles" still exist along the west of the Arctic region, showing large uncertainties in SIT prediction in these areas.

The issues related to how many ensemble candidates should be combined to improve the model performance and whether the results change with space are investigated in Figure 12, where the Arctic area is separated into 10 regions. The results show





that large spatial differences based on model selection exist for the Arctic regions, combining less than 30 ensemble candidates in the Lincoln Sea, Greenland Sea, and Canadian Arctic Archipelago, and greater number of candidates (over 80) in the Barents Sea, Kara Sea, Chukchi Sea, and Baffin Bay, and combining nearly 60 models for other regions. Figure 13 reflects the weight distributions of each region, showing that the roles of different ensemble candidates vary in different regions. Therefore, the multimodel superensemble structure is far more selective in its assignment of weights.

In summary, this study has incorporated an improved weight-determined algorithm in the multimodel superensemble structure to predict the SIT. A multi-criteria evaluation was used to validate the model. The study insights are summarised below.

- The AFTER algorithm can effectively avoid overfitting and instability in the conventional superensemble forecasting method, demonstrating better SIT simulation performance than that of the other mainstream ensemble forecast methods through a multi-criteria evaluation.

- Large biases in the SIT simulations between the dataset from the improved ensemble method and the observations were found along the coastline in the west and in August, which was in accordance with the largest SIT anomaly in time and space. This result is restricted by the limited simulations of internal variability and external forcing of SIT from all CMIP5 selected ensemble candidates and can be improved by further developing the GCMs.

- This method was used to forecast the September mean SIT in the next three decades, where the bias-removed ensemble

mean method was used as a control group. The results from these two methods exhibited a consistent spatial pattern of a continuous thinning trend in the west and an expanded ice-free area in the east. However, differences between these two high-performing ensemble methods still exist in the Canada Arctic Archipelago, Greenland Sea, and central Arctic Basin, which are enhanced over time.



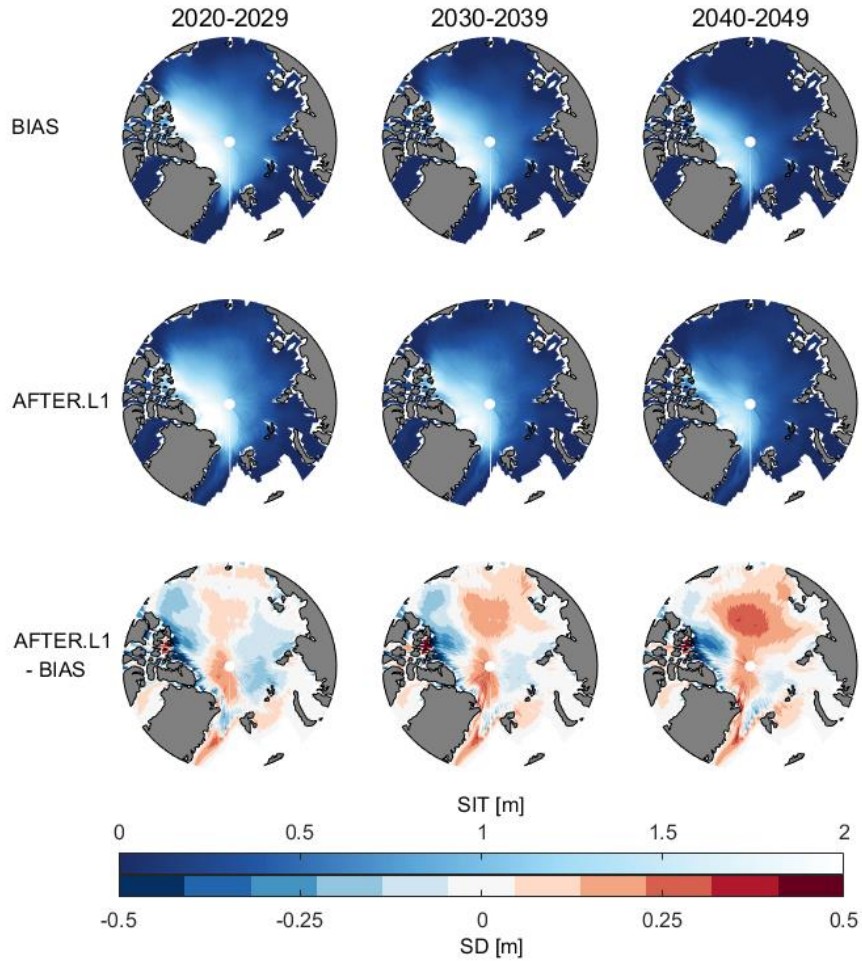

**Figure 11: Ensemble forecast of the September mean SIT derived from all the ensemble candidates listed in Table 1 during 2020–2049, using the bias-removed ensemble mean (top row) and the improved superensemble method with L1-norm AFTER (middle row). The bottom row displays the results of the AFTER.L1 – BIAS method; hence, cold colour areas are where AFTER.L1 has a reduced SIT, and the warm colour areas are where AFTER.L1 has an increased SIT.**



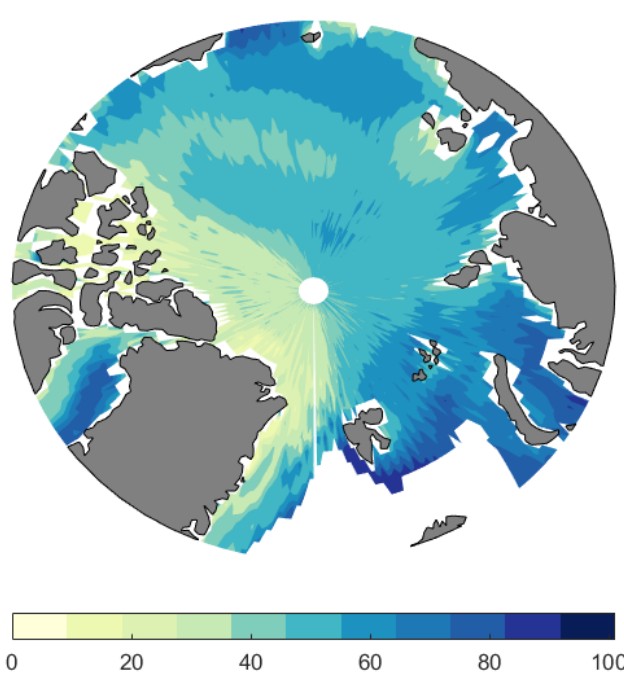

**Figure 12: Number of ensemble candidates used in the improved superensemble method with L1-norm AFTER, where 95% information can be explained. AREAS: (1) Central Arctic Ocean, (2) Lincoln and Greenland Sea, (3) Canadian Arctic Archipelago, (4) Beaufort Sea, (5) Chukchi Sea, (6) East Siberian Sea, (7) Laptev Sea, (8) Kara Sea and Barents Sea, (9) Norwegian Sea, and (10) Baffin Bay.**

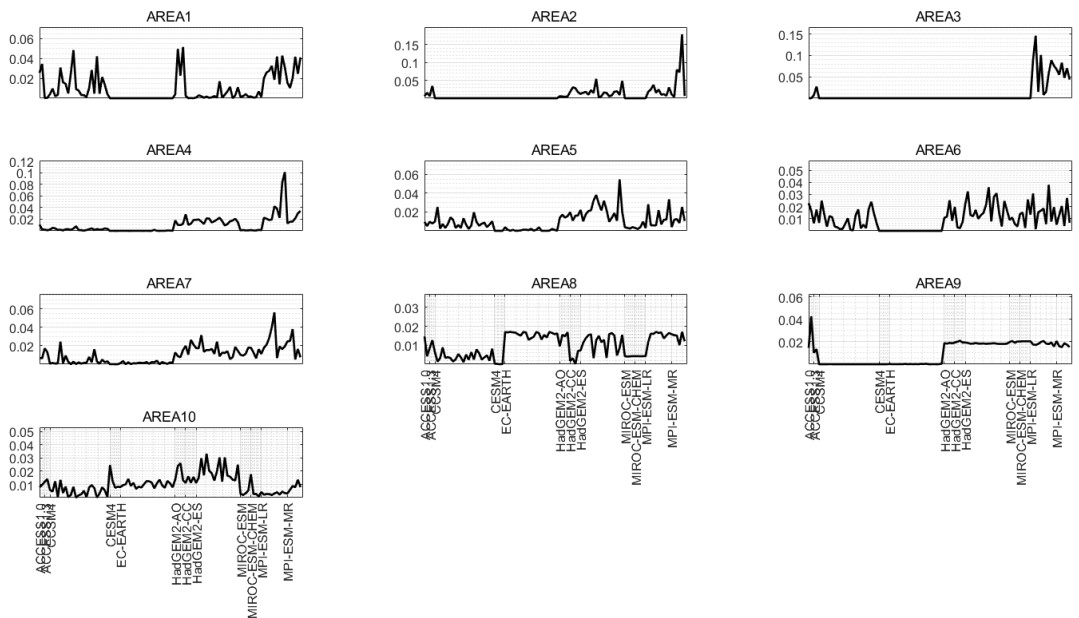

**Figure 13: Weight of each candidate model derived from the improved superensemble method with L1-norm AFTER in different regions. Areas 1–10: (1) Central Arctic Ocean, (2) Lincoln Sea and Greenland Sea, (3) Canadian Arctic Archipelago, (4) Beaufort Sea, (5) Chukchi Sea, (6) East Siberian Sea, (7) Laptev Sea, (8) Kara Sea and Barents Sea, (9) Norwegian Sea, and (10) Baffin Bay.**



**Funding:** This work has received funding from the National Natural Science Foundation of China under grant agreement number 41375002.

**Conflicts of Interest:** The authors declare no conflicts of interest.

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





**Code and Data availability**

**Code:**

Adaptive Forecasting Through Exponential Re-weighting algorithm:

Yang, Y., 2001: Combining forecasting procedures: Some theoretical results. Econom. Theory, 20, 176–222, https://doi.org/10.1017/S0266466604201086.

Structural Similarity Index Measure algorithm:

**Data:**

ACCESS1.0, ACCESS1.3 datasets:

Bi, D., and Coauthors, 2013: The ACCESS coupled model: description, control climate and evaluation. Aust. Meteorol. Oceanogr. J., https://doi.org/10.22499/2.6301.004.

CCSM4, CESM1 datasets:

Gent, P. R., and G. Danabasoglu, 2011: Response to Increasing Southern Hemisphere Winds in CCSM4. J. Clim., 24, 4992–4998.

EC-EARTH datasets:

Fichefet, T., and M. A. M. Maqueda, 1999: Modelling the influence of snow accumulation and snow-ice formation on the seasonal cycle of the Antarctic sea-ice cover. Clim. Dyn., 15, 251–268.

HadGEM2-ES, HadGEM2-CC, HadGEM2-AO datasets:

Mclaren, A. J., H. T. Banks, C. F. Durman, J. M. Gregory, and S. W. Laxon, 2006: Evaluation of the sea ice simulation in a

new coupled atmosphere-ocean climate model (HadGEM1). J. Geophys. Res., 111, C12014.

MIROC-ESM, MIROC-ESM-CHEM datasets:

Watanabe, M., M. Chikira, Y. Imada, and M. Kimoto, 2011: Convective control of ENSO simulated in MIROC. J. Clim., 24, 543–562.

MPI-ESM-LR,MPI-ESM-MR datasets:

Notz, D., F. A. Haumann, H. Haak, J. H. Jungclaus, and J. Marotzke, 2013: Arctic sea-ice evolution as modeled by Max Planck Institute for Meteorology's Earth system model. J. Adv. Model. Earth Syst., 5, 173–194.

PIOMAS datasets:

Zhang, J., and D. A. Rothrock, 2003: Modeling Global Sea Ice with a Thickness and Enthalpy Distribution Model in Generalized Curvilinear Coordinates. Mon. Weather Rev., https://doi.org/10.1175/1520-

0493(2003)131<0845:mgsiwa>2.0.co;2.



**Author contribution**

| | | |
|---|---|---|
| | Wang Yangjun: | Methodology; Software; Writing- Original draft preparation |
| | Liu Kefeng: | Writing - Reviewing and Editing |
| 470 | Zhang Ren: | Supervision; Conceptualization |
| | Qian Longxia: | Data curation; Validation |
| | Zhang Yu | Language Editing |



**Competing interests**

The authors declare that no competing interest exist.