# Peer review of "Improved Multimodel Superensemble Forecast for Sea Ice Thickness using Global Climate Models"

_The Cryosphere, 2020_

## Referee Comment (RC1) · Anonymous Referee #1 · 20 Jul 2020

The authors present the rationale for a technique for improving multimodel super-ensemble forecasts (for ice thickness) using ensembles from global climate models. The CMIP5 GCMs discussed by Wang and Overland (2015) represent the basis for this study where a total of 101 ensemble forecasts from ACCESS1.0, ACCESS1.3, CSSM4, CESM1, EC-EARTH, HadGEM2-ES, HadGEM2-CC, HadGEM2-AO, MIROC-ESM, MIROC-ESM-CHEM, MPI-ESM-LR, and MPI-ESM-MR are used. They effectively demonstrate that simply averaging the ensemble members does not provide meaningful insight. Next, they investigate a bias-removed ensemble mean method (MAVRIC) where observations from PIOMAS are utilized. A series of additional techniques are explored including artificial neural network (ANN), adaptive forecasting

through experimental re-weighting (AFTER.L1 and .L2). After performing training simulations for the period of 2006-2017, the monthly average SIT RMSE for 2018 was lowest with AFTER.L2, followed closely by AFTER.L1, and ANN. MAVRIC RMSE was significantly higher in addition to the individual (101) ensemble forecasts. The temporally averaged SIT RMSE for these techniques shows that AFTER.L1 and AFTER.L2 performed best, with high values along the northern and northeast coast of Greenland. The spatially averaged SIT CC show similar results as the RMSE, with AFTER.L2 performing best (except for Jul-Oct) followed closely by AFTER.L1 and ANN. A structure similarity index measure (SSIM) is derived and it shows that AFTER.L2 and AFTER.L1 perform best with the superensemble lagging throughout the year versus all other techniques. An EOF analysis reveals that over 99% of the cumulative total variance is found in the first four patterns. The superensemble shows no agreement to observations for any of the first four components. ANN and AFTER.L2 show the best agreement with observations. Similar trends are found in the first two PC's, while ANN was the only method to match obs for PC3. An analysis of the sea ice volume revealed that the superensemble and MAVRIC showed the poorest comparison with observations. A ranking (Table 3) shows how each ensemble method compares amongst the others when evaluating skill in RMSE, CC, SSIM, EOFs, PCs and SIV. Overall, AFTER.L2 is ranked first with AFTER.L1 second, but with the superensemble finishing third ahead of Ann (fourth). I find this to be a well written paper with a thorough description of the techniques and analysis methods used. The graphics and tables are well laid out. I find that this research will be valuable to the community. I recommend publication with minor revisions noted below.

Specific Comments

In lines 262-263, the authors state "Hence, the L1-norm AFTER superensemble method is adopted to predict future variations in September SIT". The rankings in Table 3 show that overall, AFTER.L2 has the top ranking, although overall similar to the second ranked AFTER.L1. Please comment on your selection. I assume it is because

L1 was ranked first in RMSE (space and time), CC(space and time), and SSIM. Line 38: Please provide an example of "other forecasting"? Line 81: Briefly describe the four emission scenarios.

Technical Comments:

Line 49: Yang (2001a or 2001b)? Line 84: Replace "into" with "onto" Line 85: Should 2050 be 2049? Throughout the rest of the text, the years 2019-2049 are used. Table 1: RCP's should be labeled as "2.6, 4.5, 6.0, 8.5" Line 91: Replace "they" with "the"? Is this what is meant? Line 99: Is there a reference for the Fifth Report? Line 107: Specify Melia et al. (2015a, 2015b or 2015c). Figure 1: Provide more details in caption. Table 3: Include in caption that colors of boxes represent rankings as well. Figure 7: I assume the results for AFTER.L1 lay underneath AFTER.L2 as I don't see that data represented on the plot? Figure 12: Can you add boxes that show the 10 regions discussed?

---

## Referee Comment (RC2) · Anonymous Referee #2 · 22 Sep 2020

General comments: The paper describes a new method for weighting models in a multi-model ensemble for Arctic sea ice thickness prediction. This is an important topic given the need to provide reliable predictions of changing Arctic sea ice. The authors test and compare this new method to multiple other methods which have been used previously. The various methods are demonstrated and applied to CMIP5 climate model projections with metrics for a single year (2018) used to compare them. They are then applied to projections of sea ice thickness for the 2020s, 2030s, and 2040s. Some of the metrics and analysis used would likely be relevant for initialized predictions of sea ice. However, I believe that there are some problems with this methodology and the metrics used in the paper particularly for the uninitialized climate projections to which they

are applied. These are further discussed below. To address these issues will require some major reworking of the manuscript, including a better consideration/discussion of model dependence and better metrics for assessment of the different techniques in the context of climate model projections. Given these concerns, I am recommending that the paper be rejected in its current form.

Specific comments:

1. Much of the material in the introduction is related to weather or seasonal prediction in which initialized forecasts are use. However, climate projections on longer timescales (such as those used here from the CMIP5 models) have some different considerations. In particular, the non-initialized climate model simulations (such as those in CMIP5) are not meant to predict the conditions from a single year, in which internal variability plays a large role, but instead the ensemble mean conditions represent the forced response of the system, which is relevant to longer timescale predictions. Given this, I am uncertain why metrics for a single year (2018) are used to assess the different ensemble techniques as applied to CMIP5 simulations. Could you please further explain, discuss this? Alternatively, you may want to consider assessing the methods using a perfect model setup in which you treat a model projection as the truth (for example, see Karpechko et al., 2013, doi: 10.1175/JAS-D-13-071.1) and note that in doing this, that model independence needs to be considered (see more on independence below).

2. A primary rationale for using multi-model ensemble predictions for climate is the thought that different models will have different biases and so considering multiple models allows us to characterize the uncertainty associated with imperfect models and to "average out" some of that uncertainty in our projections of the forced climate changes. Because of this, having models that are independent samples is an important consideration. Ensemble members from individual models are not independent since they are produced by the same model and ideally should only differ due to the internal variability that is not predictable on the timescales of consideration in uninitialized simulations. Because of this, in producing a simple multi-model ensemble mean (for example in

the IPCC), often only a single realization from each individual model is used. Additionally, there is considerable interdependence of models, especially from the same centers (for example see Knutti et al.s 2013 paper on climate model genealogy), which can be a problem for producing an ensemble average. It appears to me that for some of the ensemble techniques (like the simple ensemble mean), you have treated every simulation including ensemble members from an individual model as independent simulations and averaged them all together. Is this correct? If so, those methods will be heavily weighted towards the few models that have numerous members, which should be avoided. In general, the issue of model independence and the challenges inherent in the climate model ensembles of opportunity should be discussed in the manuscript and considered in the different ensemble techniques.

3. There are numerous papers discussing ensemble averaging techniques for climate projections for many different properties but these are not generally mentioned in the introduction. Many of these papers discuss challenges with model weighting in the CMIP ensembles of opportunity and some strategies behind devising reasonable weights. Please expand the introduction to include some of this background material (for a recent sea ice example see – Knutti et al., 2017, doi: 10.1002/2016GL072012).

4. Please enhance the description of the metrics used in the assessment of the different methods. Are you comparing the single time-varying multi-model ensemble values from each different method with the observations for 2018? It was not always clear whether things like the correlation coefficients were computed in space (with appropriate weighting for non-equal area grid cells) or computed in time (just over the months in 2018) or something else. Are all of the metrics just computed for 2018 using monthly data?

5. Why does the conventional multimodel superensemble method provide such bad results?

6. The observed training data that is used is only about 10 years long. On this

timescale, internal variability may have a large imprint on the trends and variability. Given this, uninitialized climate model simulation disagreement with observations may not mean that the model is deficient but could just be a consequence of internal variability. The weights will not necessarily reflect this and so "good" models could be down-weighted in the projections. Please discuss these limitations in the paper and the challenges in using a short observational record in comparison to the free-running climate simulations which have their own time-evolution of internal variability.

---

## Author Comment (AC1) · 5 Dec 2020

Dear Reviewer,

Thank you for allowing a revision of our manuscript, with an opportunity to address your comments. We are uploading our point-by-point response to the comments (below) . The read friendly version can be seen in the supplement.
* * *
Reviewer#1, Concern # 1: In lines 262-263, the authors state "Hence, the L1-norm AFTER superensemble method is adopted to predict future variations in September SIT".

[Figure]

The rankings in Table 3 show that overall, AFTER.L2 has the top ranking, although overall similar to the second ranked AFTER.L1. Please comment on your selection. I assume it is because L1 was ranked first in RMSE (space and time), CC(space and time), and SSIM.
* * *
Author response: Thanks for your kindly advice. As the results from AFTER.L1 are similar to the AFTER.L2, we adopted only one algorithm in the study, that is, AFTER.L1. The results of AFTER.L2 are not mentioned in this manuscript.
* * *
Reviewer#1, Concern # 2: Line38: Please provide an example of "other forecasting"? Line 81: Briefly describe the four emission scenarios. Line 49: Yang (2001a or 2001b)? Line 84: Replace "into" with "onto" Line 85: Should 2050 be 2049? Throughout the rest of the text, the years 2019-2049 are used. Table 1: RCP's should be labeled as "2.6, 4.5, 6.0, 8.5" Line 91: Replace "they" with "the"? Is this what is meant? Line 99: Is there a reference for the Fifth Report? Line 107: Specify Melia et al. (2015a, 2015b or 2015c). Figure 1: Provide more details in caption. Table 3: Include in caption that colors of boxes represent rankings as well. Figure 7: I assume the results for AFTER.L1 lay underneath AFTER.L2 as I don't see that data represented on the plot? Figure 12: Can you add boxes that show the 10 regions discussed?
* * *
Author response: Thanks for your advice. 1) Line 38 (new line 49), the unclear statement has been removed. 2) Line 81 (new line 89), The representative concentration pathways (RCPs), including 2.6, 4.5, 6.0, and 8.5 (van Vuuren et al., 2011), which corresponds to the radiative forcing of +2.6, +4.5, +6.0 and +4.5 Wm in 2100 relative to pre-industrial levels. 3) Yang (2001b) 4) New line 91, "into" has been replaced with "onto" 5) In this paper, 2050 has been used throughout the paper 6) New Table
1, RCPs are labeled as 2.6, 4.5, 6.0, 8.5 7) Line 91 (new line 98), "they" has been replaced with "them", representing multiple information sources 8) The reference has been added in new line 108, (Stocker et al., 2013) 9) New line 72, only one reference Melia, (2015) has been remained. 10) In order to highlight the study purpose, some ensemble methods are not mentioned in the revised paper, so as some test methods. Thus, the original Figure 1 and ranked table have been removed. 11) As the results from AFTER.L1 are similar to the AFTER.L2, we adopted only one algorithm in the study, that is, AFTER.L1. The results of AFTER.L2 are not mentioned in this manuscript. 12) The new Figure 9-10 are provided to show the results from different regions.

Please also note the supplement to this comment:
https://tc.copernicus.org/preprints/tc-2020-86/tc-2020-86-AC1-supplement.pdf
* * *

---

## Author Comment (AC2) · 5 Dec 2020

Dear Reviewer,

Thank you for allowing a revision of our manuscript, with an opportunity to address your comments. We are uploading our point-by-point response to the comments (below). The read friendly version can be seen in the supplement.

Dear Reviewer,

Thank you for allowing a revision of our manuscript, with an opportunity to address your comments. We are uploading our point-by-point response to the comments (below).

The read friendly version can be seen in the supplement. Reviewer#2, Concern # 1: Much of the material in the introduction is related to weather or seasonal prediction in which initialized forecasts are use. However, climate projections on longer timescales (such as those used here from the CMIP5 models) have some different considerations. In particular, the non-initialized climate model simulations (such as those in CMIP5) are not meant to predict the conditions from a single year, in which internal variability plays a large role, but instead the ensemble mean conditions represent the forced response of the system, which is relevant to longer timescale predictions. Given this, I am uncertain why metrics for a single year (2018) are used to assess the different ensemble techniques as applied to CMIP5 simulations. Could you please further explain, discuss this? Alternatively, you may want to consider assessing the methods using a perfect model setup in which you treat a model projection as the truth (for example, see Karpechko et al., 2013, doi: 10.1175/JAS-D-13-071.1) and note that in doing this, that model independence needs to be considered (see more on independence below). Author response: Thanks for your suggestions. The metrics for a single year (2018) are inappropriate to use to assess the different ensemble techniques as applied to CMIP5 simulations due to the impact of internal variability. Therefore, all the experiments have been redesigned in this study. First, as it is often assumed that internal variability can sufficiently neglect the Earth's climate system if the test period is beyond 30 years (Arguez and Vose, 2011), thus, we extended the time series, monthly data for from 1979 to 2014 were utilised in the training phase, and monthly data from 2015 to 2050 were used in the test phase. Second, the impact of internal variability on the proposed method, i.e. MMSE-AFTER method, has been discussed in section 4.1. Results show a good performance in depicting the future system evaluation of the SIT regardless of internal variability. Third, the pseudo-reality approach (a perfect model setup in which you treat a model projection as the truth (for example, see Karpechko et al., 2013, doi: 10.1175/JAS-D-13-071.1)) has been used with cross validation method to access the methods. Forth, the impact of model independence has been discussed in section 4.2.

Reviewer#2, Concern # 2: A primary rationale for using multi-model ensemble predictions for climate is the thought that different models will have different biases and so considering multiple models allows us to characterize the uncertainty associated with imperfect models and to "average out" some of that uncertainty in our projections of the forced climate changes. Because of this, having models that are independent samples is an important consideration. Ensemble members from individual models are not independent since they are produced by the same model and ideally should only differ due to the internal variability that is not predictable on the timescales of consideration in uninitialized simulations. Because of this, in producing a simple multi-model ensemble mean (for example in the IPCC), often only a single realization from each individual model is used. Additionally, there is considerable interdependence of models, especially from the same centers (for example see Knutti et al.s 2013 paper on climate model genealogy), which can be a problem for producing an ensemble average. It appears to me that for some of the ensemble techniques (like the simple ensemble mean), you have treated every simulation including ensemble members from an individual model as independent simulations and averaged them all together. Is this correct? If so, those methods will be heavily weighted towards the few models that have numerous members, which should be avoided. In general, the issue of model independence and the challenges inherent in the climate model ensembles of opportunity should be discussed in the manuscript and considered in the different ensemble techniques. Author response: Thanks for your suggestions First, Control experiments are designed to discuss the impact of model independence in section 4.2. In the control group that considers the impact of model interdependence, all the candidates from each group were first averaged, avoiding a few models that had numerous members highly weighted. Then, the procedure detailed in Section 3.2 was repeated. The experimental group selected in the second step must be chosen without intentional consideration of model interdependence. As listed in Table 7, the results (including CCs, RMSEs, and SSIMs) of the experimental and control groups were at the same level in all experiments, showing that the MMSE-AFTER method is capable of automatically

handling model interdependency. Thus, there is no need to deliberately consider model interdependency in the SIT ensemble forecast when the proposed method is used, thereby providing a more objective alternative compared with some current weighting schemes (e.g. Knutti et al., 2017). In addition, some models may have a common structural limitation, leading to whole biases (Knutti et al., 2017). Thus, the particular purpose of the ensemble forecast must be determined to be either an ensemble for adaptation (EFA) or an ensemble for improvement (EFI). EFA occurs when there is a model from all the candidates that can capture the true data generation process, or when other candidates can only provide redundant information relative to the former one. However, as different model candidates can provide different information, which may cause misspecifications in different ways, an EFI might occur. Herein, the MMSE scheme, which was adopted to consider the effects of local biases on the improvement of the ensemble forecast, combined with the AFTER algorithm, constructed a good link between the past and future of the system evolution of the SIT. Moreover, the new weighting scheme solves the problem of redundant information provided by model interdependency by focusing on the different information among different candidates.
* * *
Reviewer#2, Concern # 3: There are numerous papers discussing ensemble averaging techniques for climate projections for many different properties but these are not generally mentioned in the introduction. Many of these papers discuss challenges with model weighting in the CMIP ensembles of opportunity and some strategies behind devising reasonable weights. Please expand the introduction to include some of this background material (for a recent sea ice example see – Knutti et al., 2017, doi: 10.1002/2016GL072012). Author response: Thanks for your suggestion. The background material has been expanded in the Introduction. Some weighted schemes have been designed to combine forecasts (Abramowitz and Bishop, 2015; Sanderson et al., 2015), wherein the models that agree better with observations for a selected set of diagnostics are assigned higher weights and vice versa. Model interdependence is

another important factor in weighted schemes and has been discussed by Knutti et al. (2017). The interdependence of models, especially from the same centres, can be a problem for producing an ensemble average (Knutti and Sedláček, 2013). Knutti et al. (2017) adopted an evaluation of model interdependence into the design of the weight scheme, assigning less weight to the largely duplicate existing models.
* * *
Reviewer#2, Concern # 4: Please enhance the description of the metrics used in the assessment of the different methods. Are you comparing the single time-varying multi-model ensemble values from each different method with the observations for 2018? It was not always clear whether things like the correlation coefficients were computed in space (with appropriate weighting for non-equal area grid cells) or computed in time (just over the months in 2018) or something else. Are all of the metrics just computed for 2018 using monthly data? Author response: Thanks for your kindly advice. The experiments including the metrics to assess different methods have been redesigned in the study. The work flow of this study is as follow: ïĄň Step 1. Data pre-processing. To obtain a sufficient time series to overcome the influence of the internal variability on sea ice, both the historical and simulated data with different RCP scenarios were combined, thereby producing 94 model candidates with the same time series of 1979 to 2050. Meanwhile, all the candidates were interpolated into the same spatial resolution as the PIOMAS data. ïĄň Step 2. The 11 GCMs were divided into six groups, which are marked as ACCESS, CCSM4, ECEARTH, HADGEM, MIROC, and MPI. Here, one candidate was randomly chosen from one group to be considered as the "true values" while the other candidates were used as candidate forecast models. Data from 1979 to 2014 were used for the training sets, while data from 2015 to 2050 were used for the test sets. ïĄň Step 3. The weights of the MMSE and MMSE-AFTER methods were obtained by comparing the "true value" and the reconstructed historical data, respectively, during the training phase. ïĄň Step 4. The test sets were weighted by the obtained weights in Step 3, and then the combined forecasts were produced. ïĄň

Step 5. A candidate from a different group was selected, repeating Steps 2 – 4. ïĄň Step 6. Multiple evaluation methods (CC, RMSE, and SSIM) were adopted to measure the performance of each method in the SIT ensemble forecast from time and space. Then comes the best ensemble forecast method. ïĄň Step 7. Emergent constraints experiments have been designed to study the impact of internal variability on the best ensemble forecast method. ïĄň Step 8. Control experiments have been designed to test the impact of model interdependency on the best ensemble forecast method. ïĄň Step 9. Future variability of SIT and the best combined weights of different Arctic zones are provided by the best ensemble forecast method. Overall, The work flow of this study is demonstrated as follow (see Figure 1) :
* * *
Reviewer#2, Concern # 5: Why does the conventional multimodel superensemble method provide such bad results? Author response: In each experiment, the most similar spatial structure of the SIT to the "true values" was obtained using the conventional multimodel superensemble method during the training period. However, Values with a poorer performance than those of the other two methods were used as the trained weights in the testing period. In other words, the MMSE method cannot capture the link between the past and future evolution of the system (Notz, 2015) due to the overfitting of the MMSE method during the training period.
* * *
Reviewer#2, Concern # 6: The observed training data that is used is only about 10 years long. On timescale, internal variability may have a large imprint on the trends and variability. Given this, uninitialized climate model simulation disagreement with observations may not mean that the model is deficient but could just be a consequence of internal variability. The weights will not necessarily reflect this and so "good" models could be down-weighted in the projections. Please discuss these limitations in the paper and the challenges in using a short observational record in comparison to

the free-running climate simulations which have their own time-evolution of internal variability. Author response: Thanks for your kindly advice. As your request, we extended the time series, monthly data for from 1979 to 2014 were utilised in the training phase, and monthly data from 2015 to 2050 were used in the test phase. Then, our study adopted a new weighting scheme, called the MMSE-AFTER method, to improve the performance of ensemble forecasts in SIT simulation, wherein local biases are considered by the MMSE scheme. The AFTER algorithm was incorporated to overcome linear overfitting, which exists in the conventional MMSE method, allowing us to construct a good link between the past and future evolution system of SIT. Herein, our experiments, which were based on the pseudo-reality approach, were designed to verify our model's accuracy. Results from two additional control trials showed that our proposed method eliminated the impacts of internal variability and model interdependence. The ensemble forecast of future SIT changes provided by the MMSE-AFTER method can be applied to investigating the feasibility of sailing route opening and resource exploitation in the future. Moreover, this method can be used not only for SIT but also for other variables. More work will be done to verify this method in the future. In addition, new models from CMIP6, which has a better understanding of system evolution, can be used to test the performance of this new scheme in future research.

Please also note the supplement to this comment:
https://tc.copernicus.org/preprints/tc-2020-86/tc-2020-86-AC2-supplement.pdf